# Determining the priority semen characteristics and appropriate age for genetic improvement in Thai native roosters

**Jiraporn Juiputta[1], Wipas Loengbudnark[1], Thirawat Koedkanmark[1], Vibuntita Chankitisakul[1,2], Wuttigrai Boonkum [1,2]***

1 Department of Animal Science, Faculty of Agriculture, Khon Kaen University, Khon Kaen, Thailand,
2 The Research and Development Network Center of Animal Breeding and Omics, Khon Kaen University, Khon Kaen, Thailand

* wuttbo@kku.ac.th

## Abstract

Semen characteristics are crucial indicators of reproductive success and directly influence the production efficiency of native chickens, which play a vital role in sustainable poultry production systems in Thailand. Key semen traits, including mass movement, semen pH, semen volume, sperm viability, sperm abnormalities, and sperm concentration, are routinely evaluated to assess the reproductive performance of Thai native grandparent roosters. Improved reproductive efficiency in these chickens can lead to increased fertility rates, better hatchability, and enhanced productivity, significantly benefiting smallholder farmers and the poultry industry. To enhance the accuracy and efficiency of evaluating the genetic potential of grandparent roosters, this study aimed to compare rooster age groups, estimate genetic parameters and breeding values, and develop an optimal selection index for semen traits. A total of 3,475 test-day records of six semen traits from Thai native grandparent roosters (Pradu Hang Dum), obtained from the Khon Kaen University native chicken experimental farm, were analyzed using a multi-trait animal model with average information restricted maximum likelihood (AI-REML). The heritability estimates for semen volume, mass movement, sperm concentration, sperm abnormalities, sperm viability, and semen pH across age groups were low, ranging from 0.128 to 0.161, 0.102 to 0.115, 0.101 to 0.111, 0.090 to 0.091, 0.067 to 0.083, and 0.043 to 0.057, respectively. Repeatability estimates ranged from low to moderate, between 0.119 and 0.384 for all traits and age groups. Genetic correlations among semen traits ranged from –0.332 to 0.580, –0.554 to 0.640, and –0.631 to 0.773 for rooster age Groups 1, 2, and 3, respectively. Based on heritability and genetic correlations, the three most important semen traits identified were semen volume, mass movement, and sperm concentration. The study revealed a strong relationship between rooster age and semen quality, with the highest selection index values observed in the youngest age group (32–52 weeks). This finding highlights the potential for genetic improvement by implementing a selective breeding program targeting the top 10% of young roosters based on the developed selection index. Such a strategy would significantly accelerate genetic progress in semen traits, improving reproductive efficiency and boosting the economic viability of native chicken production. Furthermore,

**Data availability statement:** The data that support the findings of this study are available within the manuscript itself.

**Funding:** This research was funded by the Graduate School of Khon Kaen University through the research fund for supporting lecturers to admit high-potential students in their expert programs (grant number: 651T102) awarded to W.L., and by the Network Center for Animal Breeding and Omics Research, Khon Kaen University awarded to W.B.

**Competing interests:** The authors have declared that no competing interests exist.

these findings contribute to a deeper understanding of genetic evaluation in native chicken populations and can serve as a model for developing sustainable breeding programs for other poultry breeds under tropical conditions.

## Introduction

Native chickens are essential to rural economies worldwide and are considered a genetic resource for developing high-yielding strains adaptable to diverse environmental conditions [1,2]. Thai native chickens, especially Pradu Hang Dum, are very popular among consumers because their meat has a unique texture. It is an inexpensive source of high-quality protein, has good taste and texture, is low in fat, and is also healthier than commercial chickens [3,4]. Thus, conserving the outstanding genetics mentioned above is an important objective to maintain and develop production to meet the needs in terms of quantity and quality of chicken breeds. A few years ago, the performance improvement of native chickens was successful in terms of growth and production efficiency [5–7]. However, greater yield improvements reduce the reproductive ability of breeders [8]. Reproductive efficiency is crucial to farm profitability. Good reproductive performance allows continued offspring generation for economic benefit, herd replenishment, and genetic expansion after selective breeding [9]. This balance between productivity and reproductive viability underscores the importance of strategically integrating genetic conservation with selective breeding programs to secure future food resources and support resilient agricultural systems globally. Artificial insemination (AI) was employed to enhance poultry's reproductive capabilities [10]. AI facilitates the mating of individual chickens who would otherwise be incompatible due to differences in size and weight between males and females. Moreover, AI allows for precise control over the genetic characteristics of subsequent generations. In the great-grandparent (GGP) and grandparent (GP) generations, breeders select roosters with superior genetics for traits such as meat quality, disease resistance, growth rate, and genes influencing egg production in their female offspring [11–13]. This targeted selection accelerates genetic improvement within the flock. However, in contrast to commercial breeds, the Thai native breed exhibits lower egg production, resulting in fewer offspring per hen even with artificial insemination [14]. This lower reproductive rate contributes significantly to higher production costs per offspring. The quality of rooster semen is one factor that affects breeding efficiency [15]. Inferior semen quality is associated with reduced fertility and increased embryo mortality [16]. Conversely, strong positive correlations exist between rooster fertility and semen characteristics, notably sperm motility and ATP content (r = 0.76–0.82) [17,18]. High sperm concentration and semen volume are also critical factors influencing fertility. This is supported by studies in both commercial and native chickens, demonstrating a positive correlation between sperm concentration and fertility [19,20]. A high sperm viability rate, such as the 85.82% observed in guinea fowl, further indicates optimal fertility conditions [21]. Therefore, assessing the semen quality before using AI is crucial. Regular semen assessments also help in the selection of high-quality males for breeding programs, ensuring that only genetically superior roosters contribute to the next generation [21]. Numerous indicators, including ejaculation volume, semen pH, sperm concentration, sperm motility, sperm viability, and sperm abnormalities, are used to evaluate semen quality [22]. In addition, male semen quality is highly variable depending on the individual broiler, breed, season, management system [23], and age [24]. In almost all breeder flocks, the quality of reproductive capacity declines as the rooster ages. Roosters are, therefore, used for breeding less than a year [25]. Moreover, on rural backyard farms, Thai native roosters are generally raised for a longer duration than is typical [26]. However, no studies have yet investigated the potential of using aged roosters for breeding or the effect of age on breeding efficiency. Consequently, this study aimed to investigate the optimal age

range for selecting and improving animal breeds to ensure high semen quality. These findings will serve as valuable guidelines for selecting future breeders.

Currently, there are several methods available for improving semen quality in male chickens, such as environmental adjustment and management strategies [27–29]. However, these approaches often yield only short-term benefits. One option that provides lasting results is genetic selection methods for enhancing semen quality offer the potential for more enduring improvements. Most studies related to semen quality traits have focused primarily on genetic parameter estimation [30–32], which provides valuable knowledge to breeders. However, it cannot be used to individually select male chickens based on superior genetics. In addition, there are many characteristics of semen quality. Therefore, there needs to be a clear principle when selecting necessary characteristics. Planning breeding strategies is essential for planning the production of replacement sires within the herd, especially in GGP and GP flocks. There are few studies on the genetic improvement of semen quality in native chickens, and there are no studies on this topic in the Pradu Hang Dum. Despite the importance of the quality of semen used in artificial insemination for ensuring the reproductive success of poultry production, few studies have estimated the extent of genetic variability in semen quality traits, especially in native chickens.

This study was designed to estimate genetic parameters and breeding values and to develop a suitable selection index for semen traits in native Thai grandparent roosters by age. The findings from this research can guide the genetic improvement of native chickens. Additionally, they can enhance the development of genetic characteristics related to semen quality in male chickens more precisely and rapidly. Consequently, farmers may benefit from having male chickens with superior genetic reproductive performance.

## Materials and methods

The Institutional Animal Care and Use Committee approved the experimental procedures based on the Ethics of Animal Experimentation Guidelines of the National Research Council of Thailand (No. IACUC-KKU-82/66).

### Animal samples and management

This study analyzed 3,475 test-day records from six semen traits from 242 Thai native (Pradu Hang Dum) grandparent roosters at the Network Center for Animal Breeding and Omics Research (**NCAB**), Faculty of Agriculture, Khon Kaen University, Thailand. To assess the effect of age on semen quality, the roosters were divided into three age groups based on the age at first semen collection: Group 1 (32–52 weeks; n = 106), Group 2 (53–104 weeks; n = 62), and Group 3 (105–156 weeks; n = 74). A three-generation pedigree was used to select roosters and minimize genetic relatedness, thereby reducing the risk of inbreeding.

All roosters were raised under uniform conditions in an open-housing system, housed individually in cages (45 × 50 × 60 cm³), and maintained on a consistent diet (commercial feed: 90.07% dry matter, 17.15% crude protein, 3.35% crude fiber, 3.99% ether extract, 9.75% ash) with *ad libitum* water access. A 12-hour light cycle was maintained. Semen collection training commenced at 30 weeks of age.

### Data collection

Semen was routinely collected once a week using the dorsal-abdominal massage method [10] in a 1.5 mL Eppendorf tube containing 0.1 mL of IGGKPh diluent [33]. The semen samples were protected from light and kept at 22–25 °C during transport to the laboratory within 20 min after collection for macroscopic and microscopic evaluation. Semen collection was always performed by the same person to maximize semen quality and quantity, and the

semen was handled carefully to prevent cross-contamination during semen collection. The quality of fresh semen, in terms of mass movement, pH, volume, concentration viability, and abnormalities, was evaluated as described by Authaida et al. [24]. The waves of sperm movement were scored on a scale of 1–5 (1 = no sperm movement or very slow, less than 10%; 5 = very rapid waves and whirlpools visible, with more than 90% of the sperm showing forward movement). The pH of the semen was measured using a susceptible p-Hydrion test paper (the pH ranged from 6.4 to 8.0), and the results were compared to those obtained with a color chart meter. The semen volume was measured with the use of a syringe (1 ml). The sperm concentration was determined using a hemocytometer counting chamber and is expressed as billion ($10^9$) sperm cells/mL. The eosin-nigrosine staining technique was used to evaluate sperm viability and sperm abnormalities [34]. The results are reported as percentages, with stained sperm being treated as dead sperm and unstained sperm as living sperm for sperm viability and sperm abnormalities determining whether one has morphological anomalies (such as anomalies in the head, tail, connecting piece, or terminal piece) [35].

## Statistical analysis and genetic model

The PROC UNIVARIATE package in SAS v 9.0 software was used to check the quality of the data, including the data distribution, data anomalies, and descriptive statistics. The sperm viability and abnormality data were transformed by arcsine transformation before analysis. The recorded data were compared by rooster age group using a general linear model for unbalanced data (PROC GLM) with the SAS package to investigate the significance of the difference. If significant differences were detected, multiple pairwise comparisons were conducted using *Scheffe's test* ($p < 0.05$). The variance components and genetic parameters, such as heritability, repeatability, genetic correlation, and phenotypic correlations, were estimated using the average information restricted maximum likelihood (AI-REML) approach; moreover, the breeding values were estimated using the BLUPF90 family programs [36]. The model used for analysis was as follows:

### Multi-trait animal model:

$$Y = Xb + Za + Wpe + e$$

where Y is the vector corresponding to the semen quality traits; X is the incidence matrix related to fixed effects; Z and W are incidence matrices related to random effects; b is the vector of fixed effects, including the chicken hatch set, the temperature humidity index and the covariance between the chicken age group and body weight group; a is the vector of random additive genetic effects, assumed to be $a \sim N\left(0, A\sigma_a^2\right)$, where $A$ is the numerator relationship matrix and $\sigma_a^2$ is the additive genetic variance; pe is the vector of random permanent environmental effects, assumed to be $pe \sim N\left(0, I\sigma_{pe}^2\right)$, where $I$ is the identity matrix and $\sigma_{pe}^2$ is the permanent environmental variance; and e is the vector of random residual effects, assumed to be $e \sim N\left(0, I\sigma_e^2\right)$, where $\sigma_e^2$ is the residual variance.

The covariance matrices for models were:

$$\text{var}\begin{bmatrix} a \\ pe \\ e \end{bmatrix} = \begin{bmatrix} G \otimes A & 0 & 0 \\ 0 & P \otimes I & 0 \\ 0 & 0 & R \end{bmatrix}$$

where, G and P are 6 × 6 matrices of (co)variances for additive genetic and permanent environmental effects, respectively. A is the additive genetic relationship matrix among animals, I

is an identity matrix, $\otimes$ is Kronecker product between matrices, and R is the diagonal matrix of residual variances matrices corresponding to each trait. Genetic parameters, including heritability, repeatability, and genetic and phenotypic correlations, were estimated using a multi-trait animal model.

## Constructing the selection index

The selection indices (I) were calculated based on the estimated breeding value (EBV) of semen traits. The appropriate semen characteristics used in the selection index will be determined by the traits with the highest heritability and genetic correlation values. The relative economic value (v) of each semen trait was calculated as a proportion of the standardized economic value to the total economic importance of all traits undergoing genetic evaluation, with values ranging from 0 to 1. Determining the relative economic value of each semen characteristic in a selection index is crucial in breeding programs as it helps balance multiple traits according to their economic importance. Given these values, breeders can make informed decisions based on the heritability of the traits, genetic correlations between traits, and the economic importance of traits that enhance overall profitability and efficiency in breeding programs. The selection index equation is shown as follows:

$$I = \left(v_1 \times EBV_{trait1}\right) + \left(v_2 \times EBV_{trait2}\right) + \left(v_3 \times EBV_{trait3}\right)$$

where I is the selection index; $v_1, v_2, v_3$ are relative economic values (0–1) for semen quality traits; and $EBV_{trait1}, EBV_{trait2}, EBV_{trait3}$ are estimated breeding values for the traits.

## Results

### Comparisons of semen trait distributions according to age group

Comparisons of the six semen characteristics according to rooster age group are presented in Table 1. The results showed that all semen traits were significantly different according to rooster age group ($p < 0.05$). Among the rooster in group 1 (rooster age ranged from 32–52 weeks), mass movement ($3.97 \pm 0.07$), semen pH ($6.88 \pm 0.02$), semen volume ($0.39 \pm 0.01$ mL/ejaculation), sperm viability ($92.83 \pm 0.37\%$), and sperm concentration ($3.93 \pm 0.09 \times 10^9$ sperm/mL) were greater than those in Group 2 (rooster age ranged from 53–104 weeks) and Group 3 (rooster age ranged from 105–156 weeks) ($p < 0.05$); moreover, sperm abnormalities ($8.34 \pm 0.18\%$) were the lowest. The Group 3 roosters had the lowest values for semen characteristics ($p < 0.05$).

**Table 1. Least squares mean (±SE) comparisons of semen traits in Thai native roosters.**

| Parameters | Rooster age groups | | | p-value |
|---|---|---|---|---|
| | Group 1 | Group 2 | Group 3 | |
| Mass movement (score) | $3.97 \pm 0.07^a$ | $3.76 \pm 0.05^b$ | $3.46 \pm 0.09^c$ | <.0001 |
| Semen pH (score) | $6.88 \pm 0.02^a$ | $6.80 \pm 0.01^b$ | $6.76 \pm 0.02^b$ | <.0001 |
| Semen volume (mL/ejaculation) | $0.39 \pm 0.01^a$ | $0.38 \pm 0.01^a$ | $0.34 \pm 0.01^b$ | 0.0390 |
| Sperm viability (%) | $92.83 \pm 0.37^a$ | $90.02 \pm 0.25^a$ | $86.60 \pm 0.41^b$ | <.0001 |
| Sperm abnormalities (%) | $8.34 \pm 0.18^b$ | $9.56 \pm 0.19^b$ | $10.65 \pm 0.22^a$ | 0.0266 |
| Sperm concentration (×10⁹ sperm/mL) | $3.93 \pm 0.09^a$ | $3.58 \pm 0.06^c$ | $3.42 \pm 0.10^b$ | <.0001 |

Group 1 = rooster age ranged from 32–52 weeks, Group 2 = rooster age ranged from 53–104 weeks, and Group 3 = rooster age ranged from 105–156 weeks.

[a, b, c]Different superscript letters for each semen trait indicate a significant difference at $p < 0.05$.

## Variance components and genetic parameter estimates

The variance components and genetic parameters of the semen traits are presented in Table 2. All variance components and heritability values for semen traits in the rooster age groups were in the low range (less than 0.200). When sorting heritability values of each trait were from highest to lowest, semen volume (0.128–0.161) had the highest values, followed by mass movement (0.102–0.115), sperm concentration (0.101–0.111), sperm abnormalities (0.090–0.091), sperm viability (0.067–0.083), and semen pH (0.043–0.057). The semen characteristics whose heritability increased with rooster age were as follows: semen volume, sperm concentration, and semen pH. The characteristics whose values decreased with rooster age were mass movement, sperm viability, and sperm abnormalities. The additive genetic variation and permanent environmental variation increase with age.

## Genetic and phenotypic correlation estimates

Table 3 shows the genetic and phenotypic correlations between semen traits. The genetic correlations between semen traits ranged from –0.332 to 0.580, –0.554 to 0.640, and –0.631 to 0.773 for Group 1, Group 2, and Group 3 chicken roosters, respectively. Interesting genetic correlations such as semen volume vs. sperm concentration (0.561, 0.640, and 0.773 for group1, group2, and group3 roosters), mass movement vs. sperm concentration (0.439, 0.467, and 0.539 for group1, group2, and group3 roosters), and mass movement vs. semen volume (0.363, 0.382, and 0.390 for group1, group2, and group3 roosters). Phenotypic correlations

**Table 2. Estimated variance components, heritability, and repeatability for semen traits in Thai native roosters.**

| Rooster age groups | Traits | Variance components/Heritability/Repeatability | | | | |
|---|---|---|---|---|---|---|
| | | $\sigma_a^2$ | $\sigma_{pe}^2$ | $\sigma_e^2$ | $h^2\left(\pm\text{SE}\right)$ | $t\left(\pm\text{SE}\right)$ |
| Group1 | Mass movement | 0.148 | 0.192 | 0.945 | 0.115±0.02 | 0.265±0.04 |
| | Semen pH | 0.004 | 0.016 | 0.074 | 0.043±0.01 | 0.213±0.04 |
| | Semen volume | 0.005 | 0.007 | 0.027 | 0.128±0.02 | 0.308±0.04 |
| | Sperm viability | 0.001 | 0.001 | 0.010 | 0.083±0.01 | 0.167±0.03 |
| | Sperm abnormalities | 0.000 | 0.001 | 0.003 | 0.091±0.02 | 0.318±0.05 |
| | Sperm concentration | 0.144 | 0.330 | 0.954 | 0.101±0.01 | 0.332±0.05 |
| Group2 | Mass movement | 0.156 | 0.199 | 1.098 | 0.107±0.01 | 0.244±0.03 |
| | Semen pH | 0.003 | 0.011 | 0.053 | 0.045±0.01 | 0.209±0.03 |
| | Semen volume | 0.005 | 0.010 | 0.021 | 0.139±0.01 | 0.417±0.05 |
| | Sperm viability | 0.001 | 0.002 | 0.010 | 0.077±0.01 | 0.231±0.03 |
| | Sperm abnormalities | 0.001 | 0.001 | 0.005 | 0.090±0.01 | 0.242±0.03 |
| | Sperm concentration | 0.164 | 0.352 | 0.968 | 0.111±0.01 | 0.348±0.05 |
| Group3 | Mass movement | 0.165 | 0.298 | 1.151 | 0.102±0.01 | 0.287±0.03 |
| | Semen pH | 0.005 | 0.033 | 0.049 | 0.057±0.01 | 0.437±0.06 |
| | Semen volume | 0.005 | 0.011 | 0.015 | 0.161±0.03 | 0.516±0.06 |
| | Sperm viability | 0.001 | 0.004 | 0.010 | 0.067±0.01 | 0.333±0.04 |
| | Sperm abnormalities | 0.001 | 0.004 | 0.005 | 0.090±0.03 | 0.495±0.06 |
| | Sperm concentration | 0.202 | 0.398 | 1.233 | 0.110±0.01 | 0.327±0.05 |

Group 1 = rooster age ranged from 32–52 weeks, Group 2 = rooster age ranging from 53–104 weeks, and Group 3 = rooster age ranging from 105–156 weeks.

$\sigma_a^2$ = additive genetic variance, $\sigma_{pe}^2$ = permanent environmental variance, $\sigma_e^2$ = residual variance, $h^2\left(\pm\text{SE}\right)$ = heritability ( ± standard error), $t$ = repeatability ( ± standard error).

were lower than genetic correlations, ranging from −0.260 to 0.345, −0.350 to 0.330, and −0.390 to 0.360 for Group 1, Group 2, and Group 3 roosters, respectively.

## Selection indices

Fig 1 presents the separated and combined top 10%, 20%, and 30% of selection indices from the first three priority semen traits (semen volume, mass movement, and sperm concentration) according to the rooster age group. The selection indices (I) were calculated based on the estimated breeding value (EBV) of semen traits. The relative economic value (v) of each semen trait was calculated as a proportion of the standardized economic value to the total economic importance of all traits undergoing genetic evaluation, with values ranging from 0 to 1. Determining the relative economic value of each semen characteristic in a selection index is a crucial component in breeding programs. This helps balance multiple traits according to their economic importance. Given these values, breeders can make informed decisions based on the heritability of the traits, genetic correlations between traits, and the economic importance of traits that enhance overall profitability and efficiency in breeding programs. To understand the economic configuration guidelines of this study, we explain why the relative economic values were defined as 0.5 for semen volume, 0.4 for sperm concentration, and 0.1 for mass movement. Considering the heritability rate, we found that the heritability of semen volume was higher compared to sperm concentration and mass movement traits. Therefore, the economic value assigned to semen volume is higher than for the other two traits. We also considered correlation values. The genetic correlations between semen volume and sperm concentration in each age group of a rooster were the highest compared to the genetic correlations between mass movement and semen volume, as well as between mass movement and sperm concentration. This indicates that the genetic relationship between semen volume and

**Table 3. Genetic correlations (above diagonal) and phenotypic correlations (below diagonal) of semen traits in Thai native roosters.**

| Rooster age groups | Traits | Mass movement | Semen pH | Semen volume | Sperm viability | Sperm abnormalities | Sperm concentration |
|---|---|---|---|---|---|---|---|
| Group1 | **Mass movement** | – | 0.277 | 0.363 | 0.365 | −0.297 | 0.439 |
| | **Semen pH** | 0.120 | – | 0.492 | 0.015 | −0.262 | −0.146 |
| | **Semen volume** | 0.245 | 0.200 | – | 0.112 | 0.475 | 0.561 |
| | **Sperm viability** | 0.140 | 0.050 | 0.080 | – | −0.332 | 0.201 |
| | **Sperm abnormalities** | −0.020 | −0.040 | 0.010 | −0.260 | – | 0.580 |
| | **Sperm concentration** | 0.170 | −0.110 | 0.160 | 0.280 | 0.210 | – |
| Group2 | **Mass movement** | – | 0.272 | 0.382 | 0.340 | −0.218 | 0.467 |
| | **Semen pH** | 0.150 | – | 0.546 | 0.084 | −0.073 | −0.042 |
| | **Semen volume** | 0.280 | 0.140 | – | 0.119 | 0.582 | 0.640 |
| | **Sperm viability** | 0.200 | 0.010 | 0.010 | – | −0.554 | 0.183 |
| | **Sperm abnormalities** | −0.125 | −0.100 | 0.090 | −0.350 | – | 0.624 |
| | **Sperm concentration** | 0.130 | −0.150 | 0.060 | 0.140 | 0.330 | – |
| Group3 | **Mass movement** | – | 0.291 | 0.390 | 0.164 | −0.194 | 0.539 |
| | **Semen pH** | 0.180 | – | 0.315 | 0.005 | −0.071 | −0.039 |
| | **Semen volume** | 0.330 | 0.130 | – | −0.082 | 0.652 | 0.773 |
| | **Sperm viability** | 0.050 | 0.080 | 0.080 | – | −0.631 | 0.170 |
| | **Sperm abnormalities** | −0.190 | −0.030 | 0.050 | −0.390 | – | 0.645 |
| | **Sperm concentration** | 0.190 | −0.110 | 0.100 | 0.090 | 0.360 | – |

In Group 1, the rooster age ranged from 32–52 weeks; in Group 2, the rooster age ranged from 53–104 weeks; and in Group 3, the rooster age ranged from 105–156 weeks.

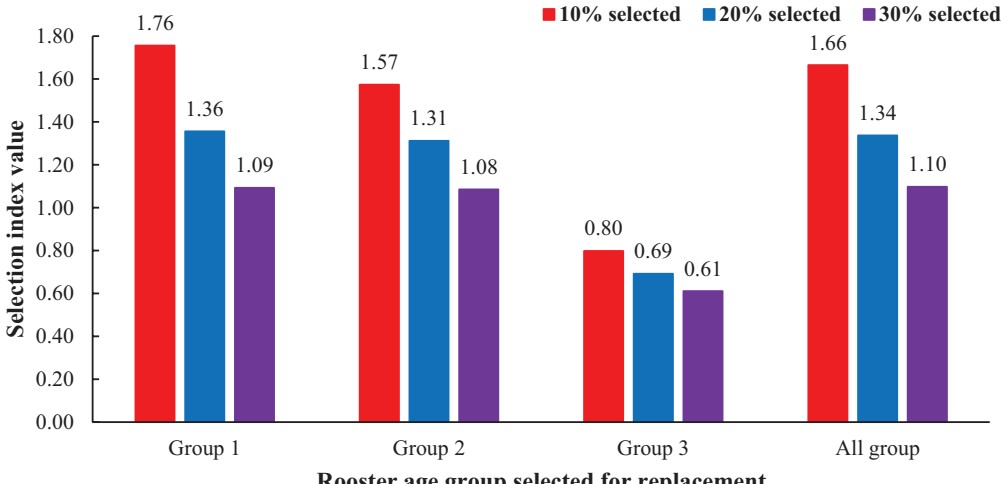

**Fig 1. Top 10%, 20%, and 30%, and averages of the selection indices values from three semen traits, separately and combined by age group.** In Group 1, the rooster age ranged from 32–52 weeks; in Group 2, the rooster age ranged from 53–104 weeks; and in Group 3, the rooster age ranged from 105–156 weeks.

sperm concentration is almost twice as strong as that between mass movement and the other traits. Therefore, the economic values of the two closely related traits, semen volume, and sperm concentration, are set to be similar to achieve genetic advancement, with the economic value of semen volume given slightly higher weight than that of sperm concentration. For general poultry breeders and producers, improving semen volume and sperm concentration should be the primary goal, followed by monitoring mass movement for quality control. Balancing these characteristics optimally leads to higher fertilization success rates, greater productivity, and better economic outcomes in poultry operations. This balanced weighting ensures that the selection process emphasizes traits that maximize economic returns while considering overall semen quality. The selection index used in the study has the following equation form:

$$I = \left(0.5 \times \mathrm{EBV}_{\text{semen volume}}\right) + \left(0.4 \times \mathrm{EBV}_{\text{sperm concentration}}\right) + \left(0.1 \times \mathrm{EBV}_{\text{mass movement}}\right)$$

The results showed that selection in the top 10% of the rooster age groups (1.76, 1.57, and 0.80 for Group 1, Group 2, and Group 3, respectively) had the highest selection indices, followed by the top 20% (1.36, 1.31, and 0.69 for Group 1, Group 2, and Group 3) and the top 30% (1.09, 1.08, and 0.61 for Group 1, Group 2, and Group 3), respectively. When data from all the rooster age groups were combined (All groups), the top 10% (1.66) had a greater selection index value than the top 20% (1.34) and top 30% (1.10). However, compared to the selection indices between separate rooster age groups and all groups, group 2 and group 3 had lower average values than the combined averages. Nonetheless, when you combine groups, you get values within the range of separate groups (lower than the highest group and higher than the lower groups).

## Discussion

Semen characteristics are important traits for poultry production and genetic improvement, especially for increasing chick production and improving the genetic potential of roosters. In

addition, semen traits are directly related to hen fertility and hatching rates. Moreover, semen traits also determine the success of poultry breeding [37]. One way to achieve this goal is by genetically improving the reproductive performance of chicken roosters. However, to genetically select male chickens, it is necessary to wait for them to reach reproductive age and continuously record the data. For this reason, considering the selection of male chickens of the appropriate age for breeding selection is an issue of interest in this study. The roosters were divided into three age groups: Group 1 had ages ranging from 32–52 weeks, Group 2 ranged from 53–104 weeks, and Group 3 ranged from 105–156 weeks. These results indicated that older chicken roosters exhibited decreased reproductive performance. This result is consistent with many past studies, including three Tanzanian native chickens (Ching'wekwe, Morogoro-medium, and Kuchi), in which the semen volume was greater in young chickens than in old chickens (p < 0.05) [38]. In addition, in Dahlem Red layer roosters in India, 42-week-old roosters had higher semen volumes and sperm concentrations, while 23-week-old roosters had the most progressive motile sperm and live sperm and the least morphologically abnormal sperm [39]. Furthermore, when determining the age of male chickens suitable for use in mating with female broiler chickens in Nigeria, it was found that male chickens at 26 weeks of age were in favorable conditions that promote optimum quality semen (sperm volume, semen color, sperm motility, semen pH, sperm concentration and live cell and semen morphological defects), which can enhance fertility and hatchability [40]. The decline in semen quality with age is associated with many physiological changes, such as decreased reproductive hormones, oxidative stress, and changes in testicular structure and function. For example, a decrease in receptors for luteinizing hormone (LH) and follicle-stimulating hormone (FSH) makes the testes less responsive to these hormones [41]. Oxidative stress, as indicated by increased malondialdehyde levels, damages the epididymal tissue [42]. In addition, aging affects the function of Sertoli cells, which are important for sperm growth and release [43]. Besides age-related factors, genetics, and environmental conditions, including health, also contribute to semen quality [44]. Given these findings, selecting young roosters for future genetic development is a more favorable approach. Younger roosters generally have a longer reproductive lifespan, allowing them to participate in breeding programs for a more extended period and produce offspring with superior traits [45]. Moreover, young roosters tend to be more resilient to environmental and management changes, which are advantageous in various production settings. Additionally, selecting young roosters enables breeders to make early decisions based on indicators of genetic potential and physical characteristics, ensuring a breeding population with desired traits. Incorporating young roosters into breeding programs not only helps prevent the decline in reproductive performance seen in older roosters but also ensures better egg hatchability [46]. This strategy improves breeding rates and provides economic benefits without significantly increasing costs [47]. Furthermore, young roosters accelerate genetic development by reducing generation time and enhancing the accuracy of genetic selection [48].

Variance components, such as additive genetic variation and permanent environmental variation, increase with the age of chickens due to the progressive expression of genetic traits, the cumulative impact of environmental factors, and the interaction between genetics and the environment over time. Studies indicate that genetic factors play a more significant role in shaping these traits as chickens mature [49]. Moreover, maternal effects, which contribute to permanent environmental variation, are more pronounced during early life but gradually decrease as chickens grow. This is because maternal influences, such as nutrition and environmental conditions, have a greater impact on young chickens but diminish as they mature [50]. This increased variation highlights the importance of considering both genetic potential and environmental management in breeding and husbandry practices. The estimated heritability

of semen traits in all the rooster age groups was low (ranging from 0.043 to 0.161; see Table 2). The values in this study were similar to those of previous studies, namely, the study of four semen characteristics (semen volume, mass movement, sperm concentration, and total sperm) in Betong chickens in Thailand. The heritability ranged from 0.04 to 0.12 [32]. For the study of semen characteristics, including sperm motility and sperm count, the heritability values were 0.08 and 0.13, respectively [31]. However, the results of this study were lower than those of many previous studies, such as in Chinese male chickens, which studied semen characteristics (semen volume, semen pH, semen color, sperm viability, sperm motility, sperm deformities, and sperm concentration), and showed that the heritability values ranged from 0.03 to 0.85 [30]. Moreover, the results of a study of seven sperm characteristics (semen color, semen volume, sperm progressive motility, sperm concentration, total sperm per ejaculate, concentration of live spermatozoa, and percentage of abnormal sperm) in Rhode Island Red and White breeder cocks in Nigeria showed heritability values ranging from 0.33 to 0.83 [51]. The low heritability in this study suggests that while genetic selection can improve semen traits, environmental factors, and management practices play a significant role in semen quality. This has shown that the presence of multiple minor alleles and epistasis contributes to the complexity of these traits, leading to low heritability. However, this study revealed that the semen characteristics of semen volume, mass movement, and sperm concentration were the three traits with the highest heritability values compared to the other traits. The results were the same for all the rooster age groups. Therefore, we recommend that animal breeders consider selecting chicken roosters at a young age (Group 1) to evaluate their genetics and select roosters to keep or discard from the flock quickly without waiting a long time to collect data, as is the case for chickens in Groups 2 and 3.

For repeatability, the values ranged from 0.167 to 0.516, 2–3 times greater than the heritability values. These results indicate that semen characteristics are affected by environmental factors rather than genetic factors. This trend is consistent across various studies previously, emphasizing that although genetic improvement is possible, effective environmental management is essential for optimizing semen quality. Environmental conditions need to be optimized to achieve the best possible expression of the trait [25,52]. Additionally, low heritability means that genetic improvements based on a single trait may be difficult to achieve. This is because the genetic component contributing to trait variation is relatively small compared to the environmental component. However, there are several strategies that animal breeders can use to achieve improvements even with low heritability, as follows. Even though the trait of interest to study has a low heritability value, a search should be performed for indicator traits that not only have a high heritability but also a strong genetic correlation to the target trait for use in genetic improvement in the future. Traits with high heritability tend to respond better to selective breeding, and focusing on traits with high heritability allows for more efficient use of resources, as efforts are directed toward traits where genetic improvement is more likely to yield positive results [53]. In addition, simultaneous genetic improvement of multiple traits, or multi-trait selection, can increase the selection accuracy for low-heritability traits. This approach takes advantage of the genetic relationships between different traits and aims to achieve overall improvements while considering complex interactions [54]. In the end, a selection index approach combines multiple traits into a single selection criterion, assigning economic weights to each trait based on its economic importance. This allows for the simultaneous improvement of several traits. The selection accuracy is increased by considering collective genetic merit rather than focusing on individual traits [55]. The genetic and phenotypic correlations between semen traits varied from negative to positive (Table 3). In general, simultaneous genetic selection for many traits should have a genetic correlation value of no less than 0.2 to take advantage of genetic linkage [56,57]. if the genes associated

with the selected traits are physically close to each other on the same chromosome, they are more likely to be inherited together [58]. In other words, a higher genetic correlation can simplify the selection process in breeding programs. If traits are strongly positively correlated, selecting one trait can indirectly improve other related traits. This reduces the complexity and cost of breeding programs. However, if traits are strongly negatively correlated, it may pose challenges for achieving genetic improvement. According to the top three semen traits with the highest heritability values (semen volume, mass movement, and sperm concentration), genetic correlations between traits will mainly consider these three traits. The genetic correlations between mass movement and semen volume, mass movement vs sperm concentration and sperm concentration vs. semen volume were moderately positive in all the rooster age groups. These values were similar to previous studies on Beijing-You chickens [30]. Genetic correlations between semen volume and sperm concentration were moderately positive, which means that genetics that control an increase in semen volume will result in an increase in sperm concentration as well. Similarly, the genetic correlations between sperm concentration and mass movement were also moderately positive, consistent with the findings of studies in Betong chickens [32], in which the genetic correlations were highly positive (0.99). A moderately positive genetic correlation in the context of genetics and heritability refers to the degree to which the genetic factors influencing one trait or characteristic are correlated with the genetic factors influencing another trait or characteristic. Specifically, individuals who share more genetic similarities for one trait are also more likely to share genetic similarities for the other trait, and the direction of the correlation is positive [59]. The selection index is therefore calculated using the EBV of the semen traits and their economic value (see Fig 1), which combines several traits of interest in an animal to estimate its overall genetic merit. It is used in selective breeding programs to decide which individuals should be used as parents for the next generation. The study results showed that the semen characteristics of young chicken grandparent roosters can be genetically selected. Moreover, semen volume, mass movement, and sperm concentration traits can be used in breeding programs simultaneously. Employing a high selection intensity, such as selecting the top 10% of roosters, maximizes the selection index by concentrating favorable alleles associated with desirable traits within the breeding population. This approach yields substantial initial genetic gains, accelerating progress [60]. However, sustained high selection intensity risks diminishing genetic diversity, potentially leading to reduced long-term response due to inbreeding depression and associated fitness consequences [61]. Strategies to mitigate this include incorporating unrelated individuals into the breeding program through periodic introduction of new sires or selection from multiple distinct genetic lines [62,63]. A balanced approach that prioritizes both short-term gains and long-term genetic health is crucial for sustainable breeding success.

This study emphasizes the importance of considering age as a critical factor in genetic evaluation models for reproductive traits. The significant decline in semen quality with increasing age underscores the need for further research into the physiological mechanisms behind age-related changes in male fertility. Additionally, these findings open opportunities for exploring strategies to extend the productive lifespan of roosters through dietary, hormonal, or environmental interventions. The study also demonstrates the utility of genetic parameters and selection indices tailored to specific age groups. By focusing on younger roosters with optimal semen quality, future research could refine breeding objectives to maximize genetic gains in reproductive traits, ensuring the long-term sustainability of native chicken populations.

From a practical perspective, the results offer valuable insights for poultry breeders and producers. Younger roosters demonstrate superior reproductive efficiency and provide a cost-effective way to enhance flock fertility and productivity. Adopting age-specific breeding

strategies, such as prioritizing roosters aged 32–52 weeks for semen collection and artificial insemination, can significantly improve fertility rates and hatchability in commercial operations. Furthermore, the study highlights the economic advantages of optimizing reproductive performance in Thai native chickens. By selectively breeding roosters with superior semen traits, producers can achieve greater genetic progress, resulting in improved flock productivity and profitability. This approach aligns with sustainable poultry production systems, particularly in smallholder farming contexts where native chickens play an essential role in household income and food security.

The economic importance of this study goes beyond the immediate advantages for smallholder farmers. Improving reproductive efficiency helps reduce input costs by minimizing the need to maintain older, less productive roosters, thereby enhancing overall production efficiency. Moreover, since Thai native chickens play a vital role in sustainable poultry production systems in tropical regions, enhancing their genetic potential supports the preservation of local breeds and biodiversity. These findings also provide a model for improving other tropical poultry breeds, demonstrating how scientific advancements can be applied to develop practical solutions that address both economic and environmental challenges [45,64,65]. In summary, this study establishes a framework for age-specific genetic improvement programs that benefit not only the poultry industry but also the scientific community. By combining genetic insights with practical breeding strategies, it offers a pathway to achieving sustainable and economically viable native chicken production systems in tropical environments.

In conclusion, to increase the genetic potential of semen traits in Thai native grandparent roosters, we suggest using a multi-trait animal model and selection index. We also conclude that the best strategy for genetic improvement to achieve rapid genetic progress is to select the top 10% of grandparent roosters at 32–52 weeks of age based on an index that includes semen volume, sperm concentration, and mass movement. Future research should investigate the persistence of fertility throughout a rooster's lifespan to optimize reproductive performance. This research should incorporate genome-wide association studies (GWAS) and hormonal analyses to identify genetic loci and physiological mechanisms affecting reproductive longevity. Integrating data on fertility traits across all age groups will allow breeding programs to target traits that enhance both early reproductive efficiency and long-term productivity, thereby maximizing genetic gain and economic returns in poultry production.

## Author contributions

**Conceptualization:** Jiraporn Juiputta, Vibuntita Chankitisakul, Wuttigrai Boonkum.

**Data curation:** Jiraporn Juiputta, Thirawat Koedkanmark, Vibuntita Chankitisakul, Wuttigrai Boonkum.

**Formal analysis:** Jiraporn Juiputta, Wipas Loengbudnark, Thirawat Koedkanmark, Wuttigrai Boonkum.

**Funding acquisition:** Wuttigrai Boonkum.

**Investigation:** Vibuntita Chankitisakul, Wuttigrai Boonkum.

**Methodology:** Jiraporn Juiputta, Wuttigrai Boonkum.

**Project administration:** Wuttigrai Boonkum.

**Supervision:** Wuttigrai Boonkum.

**Validation:** Jiraporn Juiputta, Wipas Loengbudnark, Vibuntita Chankitisakul, Wuttigrai Boonkum.

**Visualization:** Vibuntita Chankitisakul, Wuttigrai Boonkum.

**Writing – original draft:** Jiraporn Juiputta, Wipas Loengbudnark, Vibuntita Chankitisakul, Wuttigrai Boonkum.

**Writing – review & editing:** Jiraporn Juiputta, Wipas Loengbudnark, Vibuntita Chankitisakul, Wuttigrai Boonkum.

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
