## [Decision Letter · Decision Letter 0]

25 Oct 2024

PONE-D-24-30630Determining the priority semen characteristics and appropriate age for genetic improvement in Thai native roostersPLOS ONE

Dear Dr. Boonkum,

Thank you for submitting your manuscript to PLOS ONE. After careful consideration, we feel that it has merit but does not fully meet PLOS ONE’s publication criteria as it currently stands. Therefore, we invite you to submit a revised version of the manuscript that addresses the points raised during the review process.

We look forward to receiving your revised manuscript.

Kind regards,

Lamiaa Mostafa Radwan, Ph.D.

Academic Editor

PLOS ONE

Journal Requirements:

When submitting your revision, we need you to address these additional requirements. 1. Please ensure that your manuscript meets PLOS ONE's style requirements, including those for file naming. The PLOS ONE style templates can be found at  https://journals.plos.org/plosone/s/file?id=wjVg/PLOSOne_formatting_sample_main_body.pdf and https://journals.plos.org/plosone/s/file?id=ba62/PLOSOne_formatting_sample_title_authors_affiliations.pdf 2. We suggest you thoroughly copyedit your manuscript for language usage, spelling, and grammar. If you do not know anyone who can help you do this, you may wish to consider employing a professional scientific editing service.  The American Journal Experts (AJE) (https://www.aje.com/) is one such service that has extensive experience helping authors meet PLOS guidelines and can provide language editing, translation, manuscript formatting, and figure formatting to ensure your manuscript meets our submission guidelines. Please note that having the manuscript copyedited by AJE or any other editing services does not guarantee selection for peer review or acceptance for publication.  Upon resubmission, please provide the following:• The name of the colleague or the details of the professional service that edited your manuscript• A copy of your manuscript showing your changes by either highlighting them or using track changes (uploaded as a *supporting information* file)• A clean copy of the edited manuscript (uploaded as the new *manuscript* file) 3. We note that the grant information you provided in the ‘Funding Information’ and ‘Financial Disclosure’ sections do not match. When you resubmit, please ensure that you provide the correct grant numbers for the awards you received for your study in the ‘Funding Information’ section. 4. Thank you for stating the following financial disclosure:  “the graduate school Khon Kaen University through the research fund for supporting lecturers to admit high potential students to study and research on his expert program (grant number: 651T102).” Please state what role the funders took in the study. If the funders had no role, please state: "The funders had no role in study design, data collection and analysis, decision to publish, or preparation of the manuscript." If this statement is not correct you must amend it as needed.  Please include this amended Role of Funder statement in your cover letter; we will change the online submission form on your behalf. 5. Please note that funding information should not appear in the Acknowledgments section or other areas of your manuscript. We will only publish funding information present in the Funding Statement section of the online submission form. Please remove any funding-related text from the manuscript.  6. When completing the data availability statement of the submission form, you indicated that you will make your data available on acceptance. We strongly recommend all authors decide on a data sharing plan before acceptance, as the process can be lengthy and hold up publication timelines. Please note that, though access restrictions are acceptable now, your entire data will need to be made freely accessible if your manuscript is accepted for publication. This policy applies to all data except where public deposition would breach compliance with the protocol approved by your research ethics board. If you are unable to adhere to our open data policy, please kindly revise your statement to explain your reasoning and we will seek the editor's input on an exemption. Please be assured that, once you have provided your new statement, the assessment of your exemption will not hold up the peer review process.

**Additional Editor Comments:**

Dear Dr., Wuttigrai Boonkum

Thank you for submitting your manuscript to PLOS ONE. After careful consideration, we have decided that your manuscript needs Major Revision.

*Comment editor*

The manuscript needs further interpretation of the results and in-depth discussion of the data.

Kind regards,

Prof. Lamiaa Mostafa Radwan, Ph.D.

Academic Editor

PLOS ONE

Reviewer 1

The paper evaluates genetic parameters for semen quality traits in local chickens. The methodology for variance components estimation is adequate but the design is poorly described (particularly populations structure and division into groups), the rationale for choosing traits and their economic weights in index is not correct. Recommendation of 10% selected is ad hoc without considering long term response to selection. The paper needs significant improvements in both analysis and writing.

Detailed comments

Data access – there was no link or access to the data files provided

L 40 Why does genetic correlation between the semen traits instead of genetic correlation to fertility determine which traits are important?

L41-46 Not clear what the authors are trying to say

L 71. Do you mean high breeding value for egg production? Roosters don’t have egg production

L73-76 Please clarify the thought process because these seem to be unrelated statements. Is the small number of offspring produced in order to constrain inbreeding rate or because of low fertility? Can pooled semen be used to increase fertility?

L79 Which quality trait or combination of traits has the strongest association with fertility?

L87 “of age” not needed?

L90 check language

L127 The description of the design is not clear. What were the age groups if semen was collected at 32weeks? If “the data were recorded for genetic analysis only when they were 32 weeks.” where does repeatability come from? Was it the same group of roosters recorded at different ages or is there a confounding between group and age? How many roosters were in each group? What was the genetic relationship between them? How deep was the pedigree?

L197 unfinished sentence

L199-211 Not needed – these are textbook definitions of genetic parameters

The rationale for the index weights is incorrect. Economic weight should be based on how much more profit is generated by increasing that trait by one unit not by heritability and correlations between the indicator traits. If the goal trait is fertility subindex should be determined by optimizing weights to maximize fertility.

L 417-420 If young roosters have good fertility maybe selection focus should focus on improving persistency of fertility so older age records would be more relevant.

L433 indicator traits not only have to have high heritability but also strong genetic correlation to the target trait

L454 or make it more difficult if the direction is undesirable

L484 high selection intensity results in higher inbreeding rate which can reduce long term response to selection, also other traits will be included in the selection index such as growth, meat quality, feed efficiency. Expected results of alternative programs can be compared using deterministic or stochastic simulations

Reviewer 2

If the goal is to publish in a prestigious journal, it would be advisable to reconsider the analysis of the results, expand the scope of the research, perhaps by adding additional data or conducting broader comparisons with other types or breeds of poultry, and deepen the theoretical discussion.

Reviewers' comments:

Reviewer's Responses to Questions

**Comments to the Author**

1. Is the manuscript technically sound, and do the data support the conclusions?

Reviewer #1: Partly

Reviewer #2: Partly

2. Has the statistical analysis been performed appropriately and rigorously? 

Reviewer #1: Yes

Reviewer #2: Yes

3. Have the authors made all data underlying the findings in their manuscript fully available?

Reviewer #1: No

Reviewer #2: Yes

4. Is the manuscript presented in an intelligible fashion and written in standard English?

Reviewer #1: No

Reviewer #2: Yes

5. Review Comments to the Author

Reviewer #1: The paper evaluates genetic parameters for semen quality traits in local chickens. The methodology for variance components estimation is adequate but the design is poorly described (particularly populations structure and division into groups), the rationale for choosing traits and their economic weights in index is not correct. Recommendation of 10% selected is ad hoc without considering long term response to selection. The paper needs significant improvements in both analysis and writing.

Detailed comments

Data access – there was no link or access to the data files provided

L 40 Why does genetic correlation between the semen traits instead of genetic correlation to fertility determine which traits are important?

L41-46 Not clear what the authors are trying to say

L 71. Do you mean high breeding value for egg production? Roosters don’t have egg production

L73-76 Please clarify the thought process because these seem to be unrelated statements. Is the small number of offspring produced in order to constrain inbreeding rate or because of low fertility? Can pooled semen be used to increase fertility?

L79 Which quality trait or combination of traits has the strongest association with fertility?

L87 “of age” not needed?

L90 check language

L127 The description of the design is not clear. What were the age groups if semen was collected at 32weeks? If “the data were recorded for genetic analysis only when they were 32 weeks.” where does repeatability come from? Was it the same group of roosters recorded at different ages or is there a confounding between group and age? How many roosters were in each group? What was the genetic relationship between them? How deep was the pedigree?

L197 unfinished sentence

L199-211 Not needed – these are textbook definitions of genetic parameters

The rationale for the index weights is incorrect. Economic weight should be based on how much more profit is generated by increasing that trait by one unit not by heritability and correlations between the indicator traits. If the goal trait is fertility subindex should be determined by optimizing weights to maximize fertility.

L 417-420 If young roosters have good fertility maybe selection focus should focus on improving persistency of fertility so older age records would be more relevant.

L433 indicator traits not only have to have high heritability but also strong genetic correlation to the target trait

L454 or make it more difficult if the direction is undesirable

L484 high selection intensity results in higher inbreeding rate which can reduce long term response to selection, also other traits will be included in the selection index such as growth, meat quality, feed efficiency. Expected results of alternative programs can be compared using deterministic or stochastic simulations

Reviewer #2: If the goal is to publish in a prestigious journal, it would be advisable to reconsider the analysis of the results, expand the scope of the research, perhaps by adding additional data or conducting broader comparisons with other types or breeds of poultry, and deepen the theoretical discussion.

6. PLOS authors have the option to publish the peer review history of their article (what does this mean? ). If published, this will include your full peer review and any attached files.

**Do you want your identity to be public for this peer review?** For information about this choice, including consent withdrawal, please see our Privacy Policy .

Reviewer #1: No

Reviewer #2: **Yes: ** Dr. Ahmed AL-ANI

---

## [Author Response · Author response to Decision Letter 0]

11 Dec 2024

We are very grateful for the critical reading and your efforts to improve the quality of the manuscript. We hope that the manuscript in its revised form will please you. The responses to each comment are listed below.

Reviewers’ comments:

Reviewer #1: The paper evaluates genetic parameters for semen quality traits in local chickens. The methodology for variance components estimation is adequate but the design is poorly described (particularly populations structure and division into groups), the rationale for choosing traits and their economic weights in index is not correct. Recommendation of 10% selected is ad hoc without considering long term response to selection. The paper needs significant improvements in both analysis and writing.

Response: Thank you for your insightful comments. We have addressed your concerns regarding the population structure, economic weights, and selection intensity as follows:

Population structures and group division: We have added a subheading, "Animal Population Samples and Management" (lines 133-142), to provide a detailed description of the animal grouping and sampling strategy.

Economic Weights: Direct economic valuation of semen traits proved impractical given data limitations. Therefore, a simplified relative weighting approach was employed, assigning weights to traits based on heritability, genetic correlations, and estimated economic importance. This method offers a practical alternative to complex bioeconomic modeling, enabling efficient genetic selection while optimizing for both direct and correlated genetic gains

Selection Intensity (10%): Our choice of a 10% selection intensity represents a balance between achieving rapid genetic improvement and mitigating the risks of inbreeding and genetic drift. This selection rate is commonly used in similar livestock improvement programs. To minimize inbreeding, we implemented a breeding strategy incorporating careful selection of breeding pairs, rigorous monitoring of inbreeding coefficients, and a plan to introduce new, unrelated genetic material into the population after 2-3 generations. This strategy is rewritten in detail in the Discussion section, where we outline our approach to maintaining genetic diversity over the long term; See lines 489-499 in the revised MS.

Query 1: Data access – there was no link or access to the data files provided

Response 1: The datasets used and/or analyzed during the current study are available from the corresponding author on reasonable request.

Query 2: L 40 Why does genetic correlation between the semen traits instead of genetic correlation to fertility determine which traits are important?

Response 2: Direct selection for fertility is hampered by its low heritability and complex polygenic architecture. Focusing on the genetic correlations between semen traits provides a more efficient and practical alternative. This approach identifies groups of traits that influence the key physiological processes underlying fertility (e.g., spermatogenesis, sperm transport). Improving these readily measurable traits (motility, concentration, morphology) with stronger heritabilities is expected to positively impact overall fertility.

Query 3: L41-46 Not clear what the authors are trying to say

Response 3: I have revised the sections you mentioned to enhance understanding by clarifying key concepts. See lines 42-47.

Query 4: L 71. Do you mean high breeding value for egg production? Roosters don’t have egg production

Response 4: You raise a valid point. While roosters themselves do not lay eggs, they are crucial in determining the egg-laying potential of their female progeny. Line 71 refers to the indirect selection of superior egg-laying traits in female offspring through careful selection of roosters. This indirect selection is based on the rooster's breeding value for traits known to have a strong genetic correlation with egg production. We have rewritten the relevant section to clearly explain how the selection of roosters based on their breeding values for traits genetically correlated with egg production. The revised text also includes additional references to support this approach. See lines 70-79 in the revised MS.

Query 5: L73-76 Please clarify the thought process because these seem to be unrelated statements. Is the small number of offspring produced in order to constrain inbreeding rate or because of low fertility? Can pooled semen be used to increase fertility?

Response 5: Thank you for pointing out the unclear connection between offspring numbers, inbreeding, and cost. The low number of offspring per hen is primarily due to the naturally low egg production rate of the native breed, not a deliberate inbreeding control strategy or AI-related fertility issues. This lower fecundity, as explained in lines 81-83, directly contributes to higher production costs.

Query 6: L79 Which quality trait or combination of traits has the strongest association with fertility?

Response 6: We have expanded the dataset to include quality traits that have the strongest association with fertility. This is detailed in the revised manuscript, see lines 83-88.

Query 7: L87 “of age” not needed?

Response 7: We have removed the word from "of age".

Query 8: L90 check language

Response 8: We have revised the statement for clarity. See line 98-100.

Query 9: L127 The description of the design is not clear. What were the age groups if semen was collected at 32 weeks? If “the data were recorded for genetic analysis only when they were 32 weeks.” where does repeatability come from? Was it the same group of roosters recorded at different ages or is there a confounding between group and age? How many roosters were in each group? What was the genetic relationship between them? How deep was the pedigree?

Response 9: For more clearly, we have added a subheading, "Animal Population Samples and Management" (lines 133-142), to provide a detailed description of the animal grouping and sampling strategy.

Query 10: L197 unfinished sentence

Response 10: We have checked and correct the sentence. See line 207.

Query 11: L199-211 Not needed – these are textbook definitions of genetic parameters

Response 11: As suggested, the textbook definitions of genetic parameters (lines 199-211) have been removed. The revised text (lines 208-209) now simply states that these parameters—heritability, repeatability, and genetic and phenotypic correlations—were estimated using a multi-trait animal model.

Query 12: The rationale for the index weights is incorrect. Economic weight should be based on how much more profit is generated by increasing that trait by one unit not by heritability and correlations between the indicator traits. If the goal trait is fertility subindex should be determined by optimizing w eights to maximize fertility.

Response 12: We understand that determining economic values in livestock involves assigning monetary values to genetic traits based on their contribution to overall economic efficiency or profitability. However, with the current level of knowledge, accurately calculating the true monetary value of each semen trait may not be feasible. As a result, we opted for an alternative method that provides sufficient explanation of the results and can be effectively used for genetic selection of these traits. For this purpose, we selected the method of simple relative weight economic value.

Simple relative weight economic value is a simplified method for assigning economic values to traits in a breeding program. Rather than calculating precise monetary values, traits are assigned weights relative to one another based on their heritability, genetic correlations, and economic significance. These weights reflect the traits' contributions to overall profitability and efficiency in breeding programs. This approach is particularly useful when precise financial data or bioeconomic modeling is unavailable.

Additionally, this method does not require extensive data or complex modeling, making it practical for small-scale or resource-limited breeding programs. It enables breeders to make quick and effective selection decisions without complicated calculations.

In conclusion, heritability and genetic correlations play a vital role in calculating economic values and developing a reliable selection index. By considering both direct genetic improvements and correlated responses, this approach helps optimize genetic progress and achieve breeding goals efficiently.

Query 13: L 417-420 If young roosters have good fertility maybe selection focus should focus on improving persistency of fertility so older age records would be more relevant.

Response 13: Thank you for raising this point. Our study indeed highlighted the importance of persistency in genetic traits; however, our findings demonstrated that the semen characteristics of semen volume, mass movement, and sperm concentration consistently showed the highest heritability across all rooster age groups. This consistency implies that these traits are genetically stable and do not decline significantly with age relative to others.

Moreover, our results suggest that selecting roosters at a younger age (Group 1: 32–52 weeks) is most advantageous. This is due to the higher genetic and phenotypic performance observed in this age group, coupled with the opportunity for rapid genetic improvement without the need to wait for older age records. These findings advocate for prioritizing young rooster traits, as they provide both accuracy and efficiency in genetic evaluation.

While this study focuses on the genetic improvement of semen traits in younger roosters, future research could investigate fertility persistency across a rooster's lifespan, incorporating GWAS and hormonal analyses to identify loci and physiological mechanisms affecting reproductive longevity. This would ultimately improve both early and long-term productivity. We have added more details in the revised manuscript's conclusion; see lines 505-512.

Query 14: L433 indicator traits not only have to have high heritability but also strong genetic correlation to the target trait

Response 14: To address the low heritability of the target trait, the revised text (lines 440-441) emphasizes the importance of identifying indicator traits exhibiting both high heritability and a strong genetic correlation with the target trait for effective genetic improvement.

Query 15: L454 or make it more difficult if the direction is undesirable

Response 15: As suggested, the revised text (lines 460-464) now explains that strong positive correlations between traits facilitate efficient selection, whereas strong negative correlations present challenges for simultaneous improvement.

Query 16: L484 high selection intensity results in higher inbreeding rate which can reduce long term response to selection, also other traits will be included in the selection index such as growth, meat quality, and feed efficiency. Expected results of alternative programs can be compared using deterministic or stochastic simulations

Response 16: To minimize inbreeding, we implemented a breeding strategy incorporating careful selection of breeding pairs, rigorous monitoring of inbreeding coefficients, and a plan to introduce new, unrelated genetic material into the population after 2-3 generations. This strategy is rewritten in detail in the Discussion section, where we outline our approach to maintaining genetic diversity over the long term; See lines 489-499 in the revised MS.

Reviewer #2:

If the goal is to be published in a prestigious journal, it would be advisable to reconsider the analysis of the results, expand the scope of the research, perhaps by adding additional data or conducting broader comparisons with other types or breeds of poultry, and deepen the theoretical discussion.

Response: We have significantly expanded the discussion section to provide more comprehensive insights. In addition, the study identifies semen volume, mass movement, and sperm concentration as the top three traits with the highest heritability, offering breeders a clear genetic focus. This targeted approach enhances the efficiency of breeding programs by prioritizing traits that are most likely to improve reproductive performance.

Additionally, the development of a selection index based on economic weights and genetic parameters represents a sophisticated yet practical strategy for multi-trait improvement. This method ensures a balanced genetic gain across multiple semen quality traits, ultimately enhancing economic outcomes.

The findings highlight the value of genetic selection for roosters with superior reproductive performance, leading to increased fertility rates and chick production. This improvement reduces costs and enhances the profitability of poultry operations. Furthermore, by focusing on a native breed, the study supports both genetic conservation and commercial viability. This dual approach preserves valuable genetic resources while meeting the demands of the market.

---

## [Decision Letter · Decision Letter 1]

12 Jan 2025

PONE-D-24-30630R1Determining the priority semen characteristics and appropriate age for genetic improvement in Thai native roostersPLOS ONE

Dear Dr. Boonkum,

Thank you for submitting your manuscript to PLOS ONE. After careful consideration, we feel that it has merit but does not fully meet PLOS ONE’s publication criteria as it currently stands. Therefore, we invite you to submit a revised version of the manuscript that addresses the points raised during the review process.

Dear Dr., Wuttigrai Boonkum

Thank you for submitting your manuscript to PLOS ONE. After careful consideration, we have decided that your manuscript needs Minor Revision.

**Comment editor **

Additional Editor Comments:

It is necessary to show the scientific and economic importance of the manuscript in abstract and discussion of results

Kind regards,

Prof. Lamiaa Mostafa Radwan, Ph.D.

Academic Editor

PLOS ONE

**Reviewer 2**

Accept

**Reviewer 3**

2- The present article delineates a scientific study that yielded predictable and consistent results with existing literature.

3- I am does not possess sufficient information regarding the appropriateness of statistical analysis.

4-The authors have made fully available all the data underlying the findings in their article.

5-The manuscript is presented in a lucid manner and is written in standard English.

The objective of this study was to develop a suitable selection index for semen characteristics by age in native Thai grandparent roosters. The findings from this research, although predictable, may provide a framework for the genetic improvement of domestic chickens.

**Reviewer 4**

After carefully revised the Manuscript Number PONE-D-24-30630R1 under the title of "Determining the priority semen characteristics and appropriate age for genetic improvement in Thai native roosters"

I think the manuscript lakes the originality and creativity. The authors must introduce new and original ideas to the scientific community, producers, and breeders. However, the manuscript just introduces and offers traditional information and results, which are known to all scientists and breeders.

We look forward to receiving your revised manuscript.

Kind regards,

Lamiaa Mostafa Radwan, Ph.D.

Academic Editor

PLOS ONE

Journal Requirements:

Additional Editor Comments:

Dear Dr., Wuttigrai Boonkum

Thank you for submitting your manuscript to PLOS ONE. After careful consideration, we have decided that your manuscript needs Minor Revision.

Comment editor

Additional Editor Comments:

It is necessary to show the scientific and economic importance of the manuscript in abstract and discussion of results

Kind regards,

Prof. Lamiaa Mostafa Radwan, Ph.D.

Academic Editor

PLOS ONE

Reviewer 2

Accept

Reviewer 3

2- The present article delineates a scientific study that yielded predictable and consistent results with existing literature.

3- I am does not possess sufficient information regarding the appropriateness of statistical analysis.

4-The authors have made fully available all the data underlying the findings in their article.

5-The manuscript is presented in a lucid manner and is written in standard English.

The objective of this study was to develop a suitable selection index for semen characteristics by age in native Thai grandparent roosters. The findings from this research, although predictable, may provide a framework for the genetic improvement of domestic chickens.

Reviewer 4

After carefully revised the Manuscript Number PONE-D-24-30630R1 under the title of "Determining the priority semen characteristics and appropriate age for genetic improvement in Thai native roosters"

I think the manuscript lakes the originality and creativity. The authors must introduce new and original ideas to the scientific community, producers, and breeders. However, the manuscript just introduces and offers traditional information and results, which are known to all scientists and breeders.

Reviewers' comments:

Reviewer's Responses to Questions

**Comments to the Author**

1. If the authors have adequately addressed your comments raised in a previous round of review and you feel that this manuscript is now acceptable for publication, you may indicate that here to bypass the “Comments to the Author” section, enter your conflict of interest statement in the “Confidential to Editor” section, and submit your "Accept" recommendation.

Reviewer #2: (No Response)

Reviewer #3: (No Response)

Reviewer #4: All comments have been addressed

2. Is the manuscript technically sound, and do the data support the conclusions?

Reviewer #2: (No Response)

Reviewer #3: Partly

Reviewer #4: Yes

3. Has the statistical analysis been performed appropriately and rigorously? 

Reviewer #2: (No Response)

Reviewer #3: I Don't Know

Reviewer #4: Yes

4. Have the authors made all data underlying the findings in their manuscript fully available?

Reviewer #2: (No Response)

Reviewer #3: Yes

Reviewer #4: Yes

5. Is the manuscript presented in an intelligible fashion and written in standard English?

Reviewer #2: (No Response)

Reviewer #3: Yes

Reviewer #4: Yes

6. Review Comments to the Author

Reviewer #2: (No Response)

Reviewer #3: 2- The present article delineates a scientific study that yielded predictable and consistent results with existing literature.

3- I am does not possess sufficient information regarding the appropriateness of statistical analysis.

4-The authors have made fully available all the data underlying the findings in their article.

5-The manuscript is presented in a lucid manner and is written in standard English.

The objective of this study was to develop a suitable selection index for semen characteristics by age in native Thai grandparent roosters. The findings from this research, although predictable, may provide a framework for the genetic improvement of domestic chickens.

Reviewer #4: After carefully revised the Manuscript Number PONE-D-24-30630R1 under the title of "Determining the priority semen characteristics and appropriate age for genetic improvement in Thai native roosters"

I think the manuscript lakes the originality and creativity. The authors must introduce new and original ideas to the scientific community, producers, and breeders. However, the manuscript just introduces and offers traditional information and results, which are known to all scientists and breeders.

7. PLOS authors have the option to publish the peer review history of their article (what does this mean? ). If published, this will include your full peer review and any attached files.

**Do you want your identity to be public for this peer review?** For information about this choice, including consent withdrawal, please see our Privacy Policy .

Reviewer #2: **Yes: ** Ahmed Khalid

Reviewer #3: No

Reviewer #4: No

---

## [Author Response · Author response to Decision Letter 1]

23 Jan 2025

Additional Editor Comments:

Dear Dr., Wuttigrai Boonkum

Thank you for submitting your manuscript to PLOS ONE. After careful consideration, we have decided that your manuscript needs Minor Revision.

Comment editor

It is necessary to show the scientific and economic importance of the manuscript in abstract and discussion of results

Kind regards,

Prof. Lamiaa Mostafa Radwan, Ph.D.

Academic Editor

PLOS ONE

Response: We are very grateful for the critical reading and your efforts to improve the quality of the manuscript. We have revised the abstract to highlight its scientific and economic significance, as shown below and in lines 20-54. This revised abstract provides crucial insights into the genetic evaluation of semen traits in Thai native grandparent roosters, offering a foundation for targeted genetic improvement. By enhancing reproductive efficiency through selective breeding, the study not only accelerates genetic progress but also boosts the economic sustainability of native chicken production, benefiting smallholder farmers and supporting the poultry industry's long-term viability in tropical regions.

“Semen characteristics are crucial indicators of reproductive success and directly influence the production efficiency of native chickens, which play a vital role in sustainable poultry production systems in Thailand. Key semen traits, including mass movement, semen pH, semen volume, sperm viability, sperm abnormalities, and sperm concentration, are routinely evaluated to assess the reproductive performance of Thai native grandparent roosters. Improved reproductive efficiency in these chickens can lead to increased fertility rates, better hatchability, and enhanced productivity, significantly benefiting smallholder farmers and the poultry industry. To enhance the accuracy and efficiency of evaluating the genetic potential of grandparent roosters, this study aimed to compare rooster age groups, estimate genetic parameters and breeding values, and develop an optimal selection index for semen traits. A total of 3,475 test-day records of six semen traits from Thai native grandparent roosters (Pradu Hang Dum), obtained from the Khon Kaen University native chicken experimental farm, were analyzed using a multi-trait animal model with average information restricted maximum likelihood (AI-REML). The heritability estimates for semen volume, mass movement, sperm concentration, sperm abnormalities, sperm viability, and semen pH across age groups were low, ranging from 0.128 to 0.161, 0.102 to 0.115, 0.101 to 0.111, 0.090 to 0.091, 0.067 to 0.083, and 0.043 to 0.057, respectively. Repeatability estimates ranged from low to moderate, between 0.119 and 0.384 for all traits and age groups. Genetic correlations among semen traits ranged from -0.332 to 0.580, -0.554 to 0.640, and -0.631 to 0.773 for rooster age Groups 1, 2, and 3, respectively. Based on heritability and genetic correlations, the three most important semen traits identified were semen volume, mass movement, and sperm concentration. The study revealed a strong relationship between rooster age and semen quality, with the highest selection index values observed in the youngest age group (32–52 weeks). This finding highlights the potential for genetic improvement by implementing a selective breeding program targeting the top 10% of young roosters based on the developed selection index. Such a strategy would significantly accelerate genetic progress in semen traits, improving reproductive efficiency and boosting the economic viability of native chicken production. Furthermore, these findings contribute to a deeper understanding of genetic evaluation in native chicken populations and can serve as a model for developing sustainable breeding programs for other poultry breeds under tropical conditions.”

In addition, we have also revised the discussion section to include original ideas relevant to the scientific community, producers, and breeders, while highlighting the economic significance of this study. See the detail below and lines 506-544 in the revised manuscript.

“This study emphasizes the importance of considering age as a critical factor in genetic evaluation models for reproductive traits. The significant decline in semen quality with increasing age underscores the need for further research into the physiological mechanisms behind age-related changes in male fertility. Additionally, these findings open opportunities for exploring strategies to extend the productive lifespan of roosters through dietary, hormonal, or environmental interventions. The study also demonstrates the utility of genetic parameters and selection indices tailored to specific age groups. By focusing on younger roosters with optimal semen quality, future research could refine breeding objectives to maximize genetic gains in reproductive traits, ensuring the long-term sustainability of native chicken populations.

From a practical perspective, the results offer valuable insights for poultry breeders and producers. Younger roosters demonstrate superior reproductive efficiency and provide a cost-effective way to enhance flock fertility and productivity. Adopting age-specific breeding strategies, such as prioritizing roosters aged 32–52 weeks for semen collection and artificial insemination, can significantly improve fertility rates and hatchability in commercial operations. Furthermore, the study highlights the economic advantages of optimizing reproductive performance in Thai native chickens. By selectively breeding roosters with superior semen traits, producers can achieve greater genetic progress, resulting in improved flock productivity and profitability. This approach aligns with sustainable poultry production systems, particularly in smallholder farming contexts where native chickens play an essential role in household income and food security.

The economic importance of this study goes beyond the immediate advantages for smallholder farmers. Improving reproductive efficiency helps reduce input costs by minimizing the need to maintain older, less productive roosters, thereby enhancing overall production efficiency. Moreover, since Thai native chickens play a vital role in sustainable poultry production systems in tropical regions, enhancing their genetic potential supports the preservation of local breeds and biodiversity. These findings also provide a model for improving other tropical poultry breeds, demonstrating how scientific advancements can be applied to develop practical solutions that address both economic and environmental challenges (64–66). In summary, this study establishes a framework for age-specific genetic improvement programs that benefit not only the poultry industry but also the scientific community. By combining genetic insights with practical breeding strategies, it offers a pathway to achieving sustainable and economically viable native chicken production systems in tropical environments.”

Reviewers’ comments:

Reviewer 2

Accept

Response: Thank you very much for your support.

Reviewer 3

2- The present article delineates a scientific study that yielded predictable and consistent results with existing literature.

Response: We are pleased that the consistency and predictability of our results were recognized. This further supports the reliability and relevance of our research methodology.

3- I am does not possess sufficient information regarding the appropriateness of statistical analysis.

Response: We would like to emphasize that this study highlights the importance of carefully selecting suitable tools for genetic estimation to ensure accurate, reliable, and meaningful results for future research.

4-The authors have made fully available all the data underlying the findings in their article.

Response: Thank you for acknowledging that all data underlying the findings in our article have been made fully available. Ensuring data transparency and accessibility is an important aspect of our work, and we are pleased this has been recognized.

5-The manuscript is presented in a lucid manner and is written in standard English.

Response: Thank you for your positive feedback on the clarity and language of our manuscript. We are glad to hear that the manuscript is presented lucidly and meets the standard for English writing. We appreciate your acknowledgment, as we have strived to ensure the content is accessible and well-written for the readership.

The objective of this study was to develop a suitable selection index for semen characteristics by age in native Thai grandparent roosters. The findings from this research, although predictable, may provide a framework for the genetic improvement of domestic chickens.

Response: Thank you very much for your support.

Reviewer 4

After carefully revised the Manuscript Number PONE-D-24-30630R1 under the title of "Determining the priority semen characteristics and appropriate age for genetic improvement in Thai native roosters"

I think the manuscript lacks originality and creativity. The authors must introduce new and original ideas to the scientific community, producers, and breeders. However, the manuscript just introduces and offers traditional information and results, which are known to all scientists and breeders.

Response: Thank you for your insightful comments. We have also revised the discussion section to include original ideas relevant to the scientific community, producers, and breeders, while highlighting the economic significance of this study. See the detail below and lines 506-544 in the revised manuscript.

“This study emphasizes the importance of considering age as a critical factor in genetic evaluation models for reproductive traits. The significant decline in semen quality with increasing age underscores the need for further research into the physiological mechanisms behind age-related changes in male fertility. Additionally, these findings open opportunities for exploring strategies to extend the productive lifespan of roosters through dietary, hormonal, or environmental interventions. The study also demonstrates the utility of genetic parameters and selection indices tailored to specific age groups. By focusing on younger roosters with optimal semen quality, future research could refine breeding objectives to maximize genetic gains in reproductive traits, ensuring the long-term sustainability of native chicken populations.

From a practical perspective, the results offer valuable insights for poultry breeders and producers. Younger roosters demonstrate superior reproductive efficiency and provide a cost-effective way to enhance flock fertility and productivity. Adopting age-specific breeding strategies, such as prioritizing roosters aged 32–52 weeks for semen collection and artificial insemination, can significantly improve fertility rates and hatchability in commercial operations. Furthermore, the study highlights the economic advantages of optimizing reproductive performance in Thai native chickens. By selectively breeding roosters with superior semen traits, producers can achieve greater genetic progress, resulting in improved flock productivity and profitability. This approach aligns with sustainable poultry production systems, particularly in smallholder farming contexts where native chickens play an essential role in household income and food security.

The economic importance of this study goes beyond the immediate advantages for smallholder farmers. Improving reproductive efficiency helps reduce input costs by minimizing the need to maintain older, less productive roosters, thereby enhancing overall production efficiency. Moreover, since Thai native chickens play a vital role in sustainable poultry production systems in tropical regions, enhancing their genetic potential supports the preservation of local breeds and biodiversity. These findings also provide a model for improving other tropical poultry breeds, demonstrating how scientific advancements can be applied to develop practical solutions that address both economic and environmental challenges (64–66). In summary, this study establishes a framework for age-specific genetic improvement programs that benefit not only the poultry industry but also the scientific community. By combining genetic insights with practical breeding strategies, it offers a pathway to achieving sustainable and economically viable native chicken production systems in tropical environments.”

Best Regards,

Wuttigrai Boonkum

Corresponding author

---

## [Decision Letter · Decision Letter 2]

5 Feb 2025

Determining the priority semen characteristics and appropriate age for genetic improvement in Thai native roosters

PONE-D-24-30630R2

Dear Dr. Boonkum,

We’re pleased to inform you that your manuscript has been judged scientifically suitable for publication and will be formally accepted for publication once it meets all outstanding technical requirements.

Kind regards,

Lamiaa Mostafa Radwan, Ph.D.

Academic Editor

PLOS ONE

Additional Editor Comments (optional):

Accept

Reviewers' comments:

Reviewer's Responses to Questions

**Comments to the Author**

1. If the authors have adequately addressed your comments raised in a previous round of review and you feel that this manuscript is now acceptable for publication, you may indicate that here to bypass the “Comments to the Author” section, enter your conflict of interest statement in the “Confidential to Editor” section, and submit your "Accept" recommendation.

Reviewer #3: All comments have been addressed

Reviewer #4: All comments have been addressed

2. Is the manuscript technically sound, and do the data support the conclusions?

Reviewer #3: Partly

Reviewer #4: Yes

3. Has the statistical analysis been performed appropriately and rigorously? 

Reviewer #3: I Don't Know

Reviewer #4: Yes

4. Have the authors made all data underlying the findings in their manuscript fully available?

Reviewer #3: Yes

Reviewer #4: Yes

5. Is the manuscript presented in an intelligible fashion and written in standard English?

Reviewer #3: Yes

Reviewer #4: Yes

6. Review Comments to the Author

Reviewer #3: The study's findings underscore the significance of genetic composition in roosters exhibiting superior reproductive performance. Following a thorough review, the authors appear to have re-evaluated their data, expanded the scope of programming, and enhanced the theoretical discourse.

Reviewer #4: After carefully revised the Manuscript Number PONE-D-24-30630R1 under the title of

"Determining the priority semen characteristics and appropriate age for genetic

improvement in Thai native roosters"

The authors made a hard effort to address all comments, and I think the manuscript could be accepted.

7. PLOS authors have the option to publish the peer review history of their article (what does this mean? ). If published, this will include your full peer review and any attached files.

**Do you want your identity to be public for this peer review?** For information about this choice, including consent withdrawal, please see our Privacy Policy .

Reviewer #3: **Yes: ** KÜRŞAT TETİK

Reviewer #4: No

---

## [Editor Report · Acceptance letter]

PONE-D-24-30630R2

PLOS ONE

Dear Dr. Boonkum,

I'm pleased to inform you that your manuscript has been deemed suitable for publication in PLOS ONE. Congratulations! Your manuscript is now being handed over to our production team.

Kind regards,

on behalf of

Prof. Dr. Lamiaa Mostafa Radwan

Academic Editor

PLOS ONE